# Co-Occurring Methylenetetrahydrofolate Reductase (*MTHFR*) rs1801133 and rs1801131 Genotypes as Associative Genetic Modifiers of Clinical Severity in Rett Syndrome

**DOI:** 10.3390/brainsci14070624

**Published:** 2024-06-21

**Authors:** Jatinder Singh, Georgina Wilkins, Ella Goodman-Vincent, Samiya Chishti, Ruben Bonilla Guerrero, Leighton McFadden, Zvi Zahavi, Paramala Santosh

**Affiliations:** 1Department of Child and Adolescent Psychiatry, Institute of Psychiatry, Psychology and Neuroscience, King’s College London, London SE5 8AF, UK; jatinder.singh@kcl.ac.uk (J.S.); georgina.e.wilkins@kcl.ac.uk (G.W.); ella.goodman-vincent@kcl.ac.uk (E.G.-V.); samiya.chishti@slam.nhs.uk (S.C.); leighton.mcfadden@kcl.ac.uk (L.M.); 2Centre for Interventional Paediatric Psychopharmacology and Rare Diseases (CIPPRD), South London and Maudsley NHS Foundation Trust, London SE5 8AZ, UK; 3Centre for Interventional Paediatric Psychopharmacology (CIPP) Rett Centre, Institute of Psychiatry, Psychology and Neuroscience, King’s College London, London SE5 8AF, UK; 4Genetica Consulting Services, Laguna Niguel, CA 92677, USA; ruben@geneticaconsulting.com; 5Myogenes Limited, Borehamwood WD6 4PJ, UK; zahavi@myogenes.com

**Keywords:** Rett Syndrome, remethylation, MTHFR, clinical severity, pharmacogenomics

## Abstract

Aim: Remethylation disorders such as 5,10-methylenetetrahydrofolate reductase (*MTHFR*) deficiency reduce the remethylation of homocysteine to methionine. The resulting hyperhomocysteinemia can lead to serious neurological consequences and multisystem toxicity. The role of *MTHFR* genotypes has not been investigated in patients with Rett Syndrome (RTT). In this study, we sought to assess the impact of co-occurring *MTHFR* genotypes on symptom profiles in RTT. Method: Using pharmacogenomic (PGx) testing, the *MTHFR* genetic polymorphisms rs1801133 (c.665C>T mutation) and rs1801131 (c.1286A>C mutation) were determined in 65 patients (18.7 years ± 12.1 [mean ± standard deviation]) with RTT as part of routine clinical care within the Centre for Interventional Paediatric Psychopharmacology (CIPP) Rett Centre, a National and Specialist Child and Adolescent Mental Health Service (CAMHS) in the UK. The clinical severity of patients was assessed using the RTT-anchored Clinical Global Impression Scale (RTT-CGI). Results: The clinical severity symptom distribution varied between the homozygous and heterozygous *MTHFR* rs1801133 and rs1801131 genotypes. Those with the homozygous genotype had a narrower spread of severity scores across several domains (language and communication, ambulation, hand-use and eye contact clinical domains). Patients with the homozygous genotype had statistically significantly greater CGI-Severity scores than individuals with a non-homozygous *MTHFR* genotype (Z = −2.44, *p* = 0.015). When comparing the ratings of moderately impaired (4), markedly impaired (5), severely impaired (6) and extremely impaired (7), individuals with the homozygous *MTHFR* genotype were more impaired than those with the non-homozygous *MTHFR* genotype (Z = −2.06, *p* = 0.039). There was no statistically significant difference in the number of prescribed anti-epileptic drugs between the genotypes. Conclusions: Our findings show that in those with a pathogenic RTT genetic variant, co-occurring homozygotic *MTHFR* rs1801133 and rs1801131 polymorphisms may act as associative genetic modifiers of clinical severity in a subset of patients. Profiling of rs1801133 and rs1801131 in RTT may therefore be useful, especially for high-risk patients who may be at the most risk from symptom deterioration.

## 1. Introduction

The level of unmet need for managing and treating mental health problems within the Rare Disease (RD) community is high. It is impossible to overstate the devastating consequences for those affected and their families, and the significant burden placed upon patients, families, health services, and society. Rett Syndrome (RTT) is an exemplar of a paediatric RD that encompasses both neuropsychiatric and physical problems. It is one of the most recognised models of abnormal synaptic plasticity [1]. Primarily caused by mutations in the protein methyl-CpG-binding protein 2 (MeCP2), RTT has a frequency of 5 to 10 cases per 100,000 females [2]. Where there is a pathogenic variant, the *MECP2* gene is suggested to disrupt long gene expression within the brain [3], probably by altering topoisomerase IIβ activity [4]. Although the United Sates Food and Drug Administration (US, FDA) has recently approved Trofinetide in RTT therapeutics [5,6], typical treatment relies on the use of off-label medications [1], which can lead to ineffective doses being used, and may increase the risk of harm to patients and/or poor outcomes. Patients with RTT are at higher risk of adverse drug reactions (ADRs) due to the severity of the illness, co-occurring disorders, and the use of multiple medications. Indeed, in a prospective study of 18,820 adult patients in England, ADRs accounted for 6.5% of hospitalisations [7]. Polypharmacy and multimorbidity contribute to the higher incidence of ADRs [8]. In the UK, a small number of actionable pharmacogenes accounts for most prescription drugs, with 8/10 patients being exposed to at least one drug that was impacted by a variant in an actionable pharmacogene [9].

Methylenetetrahydrofolate reductase (MTHFR) deficiency involved in folate metabolism, specifically the synthesis of 5-methyl-tetrahydrofolate (5-MTHF), which is the biologically active form of folate required for the remethylation of homocysteine to methionine, has been associated with neuropsychiatric illness [10]. At present, there are about 247 single nucleotide polymorphisms (SNPs) of *MTHFR* documented in the Single Nucleotide Polymorphism Database (dbSNP) [11] and according to recent evidence, approximately 200 pathogenic *MTHFR* variants have been identified [12]. The genetic polymorphisms rs1801133 and rs1801131 are said to be important polymorphisms that have an impact on MTHFR activity [13]. While the selected genetic polymorphisms (rs1801133 and rs1801131) in this study are the only two with a clinical grade pharmacogenomics (PGx) level of evidence [14], these alleles are also related to disease risk [15,16,17,18,19,20,21]. Variants in *MTHFR* result in lower levels of 5-MTHF, and this increases homocysteine, thereby causing a dysregulated production of monoamines [22] and changes in DNA methylation [23]. In those with severe *MTHFR* deficiency, 109 mutations in 171 families have been described [24]. Severe *MTHFR* deficiency in children is characterised by hyperhomocysteinemia, hypotonia, feeding issues, failure to thrive, lethargy, apnoea, and microcephaly [24]. These findings are important in the context of RTT, especially given that others have indicated that there may also be progressive developmental delay, epilepsy, and impaired gait in patients with severe *MTHFR* deficiency [25]. These symptoms are thought to be caused in part due to reduced methylation within the brain [24]. Evidence has also suggested that the *MTHFR* c.665C>T variant increases the susceptibility to epilepsy [26]. An adolescent/adult-onset *MTHFR* deficiency with a focus on epilepsy manifesting as a neuropsychiatric illness has also been identified, and shares symptom overlap with RTT [27]. This study indicated that epilepsy occurs in about half of patients with adolescent/adult-onset *MTHFR* deficiency, and these patients have a highly variable response to antiepileptic drugs (AEDs) [27]. A few cases have also documented *MTHFR* deficiency, such as in a 14-year-old girl with progressive myoclonic epilepsy [28] and in AED-induced psychosis [29].

The UK Biobank study analysed PGx variant frequency in 487,409 subjects and showed that 99.5% of subjects had at least one actionable PGx variant [30] and the Pre-emptive Pharmacogenomic Testing for Preventing Adverse Drug Reactions (PREPARE) study further demonstrated a decrease in clinically relevant ADRs using a 12 gene PGx panel [31]. This evidence implies that there is also likely to be actionable PGx findings with potential treatment implications in the RTT patient population. However, neither the frequency nor the possibility of whether *MTHFR* rs1801133 and rs1801131 polymorphisms are genetic modifiers of clinical severity in RTT patients have been explored. Based on (I) the similarities of symptom profiles between *MTHFR* deficiency and RTT, (II) parallels in *MECP2* and *MTHFR* methylation and (III) the finding that hyperhomocysteinemia over time causes irreversible neurological damage [27], it is useful to explore profiles of rs1801133 and rs1801131 polymorphisms in RTT patients. Progressive hyperhomocysteinemia would be especially important in RTT where there is already impaired methylation [3,4,32] and timely intervention would be critical to prevent further neurological decline. Preliminary reports have also suggested that hyperhomocysteinemia may itself be epileptogenic and lead to brain atrophy [33]. The overarching aim of this study was therefore to assess if *MTHFR* rs1801133 and rs1801131 polymorphisms are genetic modifiers of clinical severity in RTT. In this study, we thus (I) explored the frequency of *MTHFR* rs1801133 and rs1801131 polymorphisms in patients with RTT, (II) examined if homozygosity of these polymorphisms affects the symptom profile of RTT patients and (III) assessed if there is a relationship between these polymorphisms and AED use in patients.

## 2. Materials and Methods

### 2.1. Study Participants

Sixty-five (65) patients with RTT and a documented PGx test for *MTHFR* as part of their routine clinical care were evaluated in this study. Genetic diagnosis of RTT was confirmed using genetic reports or medical information, and where this information was unavailable, the caregiver was consulted. A clinical diagnosis based on the information provided by Neul et al. (2010) [34] was confirmed by either a Consultant Child and Adolescent Psychiatrist or an Associate Specialist specializing in RTT or from previous medical history/case notes. Where applicable, retrospective review of medical information determined the frequencies of co-occurring neuropsychiatric diagnoses and physical illness. The following descriptive data were also collected for each patient: age, gender, and race/ethnicity. The frequency of neuropsychiatric and physical illness for some of the patients in this study has been described previously [35]. For this study, we have provided the characteristics of neuropsychiatric and physical illness for all patients with an available PGx test for *MTHFR*.

### 2.2. Determination of MTHFR Variant Frequency

#### 2.2.1. Sample Collection

After obtaining caregiver consent, PGx test kits (Myogenes, Borehamwood, UK), were posted to caregivers for sample collection. In some instances, sample collection was undertaken within the Centre for Interventional Paediatric Psychopharmacology (CIPP) Rett Centre. Samples were obtained from buccal swabs between April 2021 and January 2024. Testing was managed by Myogenes, who sent the samples for analysis to proprietary facilities such as Admera Health LLC and Innovative Gx Laboratories. Admera and Innovative Gx were two CLIA-approved and CAO-certified laboratories performing molecular testing. Both laboratories performed targeted PGx using next-generation sequencing.

#### 2.2.2. PGx Test and MTHFR Reporting

##### PGx Test

After samples were analysed, a patient-specific PGx report was generated and sent to the referring physician (PS) for evaluation. Reports were discussed with the family with relevant guidance on medication prescribing. Where necessary, PS and JS received guidance from an expert physician in PGx (RBG) for a patient-specific interpretation of the results. The PGx reports contained information on (I) current medications and medications to be considered, (II) a comprehensive drug list and (III) genotype and phenotype results. In general, the comprehensive drug list included interactions for 67 genes and 294 medications that aligned with the Clinical Pharmacogenetics Implementation Consortium (CPIC), Food and Drug Administration (FDA), Association for Molecular Pathology (AMP), College of American Pathologists (CAP), American College of Medical Genetics (ACMG), and Pharmacogenomics Knowledgebase (PharmGKB) recommendations. The Innovative Gx Laboratories PGx panel genotyped 67 genes using next-generation sequencing (NGS) and qPCR with sensitivity and specificity values of 99.9% and 99.9%, respectively. The panel contained the specific genotype results of all the 67 genes. Two PGx reports were generated from Admera Health and included gene–drug interactions from a 62-gene panel.

##### MTHFR Reporting

Each PGx report provided information on two of the most common genetic polymorphisms of *MTHFR* rs1801133 (c.665C>T mutation) and rs1801131 (c.1286A>C mutation) with a clinical grade PGx level of evidence. The assignment of *MTHFR* rs1801133 and rs1801131 polymorphisms were conducted according to the predicted effect on MTHFR activity as described (mild, partial, or reduced activity) within each individual PGx report for the 65 patients. The designations were as follows:1.Homozygous designation (*n* = 17)

Individuals with rs1801133 (c.665 C>T) and rs1801131 (c.1286 A>C) homozygous genotypes that were predicted to have reduced MTHFR activity.

2.Non-homozygous designation (*n* = 48)

Individuals with rs1801133 (c.665 C>T) and rs1801131 (c.1286 A>C) heterozygous genotypes that were predicted to have either mild or partial MTHFR activity (*n* = 41). Those predicted to have normal MTHFR activity were also assigned to this group (*n* = 7).

### 2.3. Assessment of Clinical Severity

To determine global clinical change, clinical severity was assessed using the RTT-anchored Clinical Global Impression Scale (RTT-CGI) and the following clinical domains for severity (CGI-S) were assessed: language/communication, ambulation, hand use, social [eye contact], autonomic, seizures and attentiveness [36].

### 2.4. Assessment of AEDs

The AEDs being prescribed to each patient at the time of the PGx test were collected. These findings were grouped by patients’ *MTHFR* genotype profiles. A Mann–Whitney U test was run to investigate whether the number of AEDs being taken at the time of the PGx test was statistically significant between the two *MTHFR* groups.

### 2.5. Data Extraction and Statistical Analyses

For each patient, data for *MTHFR* rs1801133 (c.665 C>T) and rs1801131 (c.1286 A>C) alleles (negative, heterozygote and homozygote) were extracted from individual PGx reports and transferred to an Excel spreadsheet. Based on CPIC and PharmGKB sources, the gene–drug information was also reviewed to gain further information on the relevance of the MTHFR phenotype on patient impact. Statistics and frequency histograms were obtained using Excel (Microsoft Excel, 2019). Due to the differing sample sizes between the homozygous population (*n* = 17) and the non-homozygous population (*n* = 48), a non-parametric test, the Mann–Whitney U test, was chosen as the analysis method (IBM SPSS Statistics 29.0.1.0).

### 2.6. Informed Consent and Ethics

Caregivers of all RTT patients in this study consented for their PGx testing to be carried out by Myogenes between April 2021 and January 2024 as part of routine clinical care. The PGx data to be collected and used for research and publication purposes received ethical approval as part of the Tailored Rett Intervention and Assessment Longitudinal (TRIAL) data warehouse by the London-Bromley Research Ethics Committee (REC reference: 15/LO/1772) on 19 October 2022 (substantial amendment number 7).

## 3. Results

### 3.1. Study Characteristics

Information for *MTHFR* rs1801133 and rs1801131 genotypes was available for 65 patients. Participant demographics are presented in Table 1A. All participants were female with a mean age (±SD) of 18.7 ± 12.1 years. In our sample population, 60 had RTT, 5 had atypical RTT, and 53 were white. Among the neuropsychiatric diagnoses, generalised anxiety disorder (45%) was the most common, followed by ASD (25%). Epilepsy (77%) was the most frequently occurring physical illness in the population, followed by gastrointestinal issues (68%) and breathing problems (52%) (Appendix A). The *MTHFR* rs1801133 and rs1801131 genotypes and their impact on MTHFR activity in the sample are presented in Table 1B. The normal, mild, partial, and reduced activity classifications for MTHFR were obtained from the PGx report for each patient.

### 3.2. Frequency of MTHFR rs1801133 and rs1801131 Genotypes in the Sample

The PGx panel reported on the two *MTHFR* genetic polymorphisms (rs1801133 and rs1801131) with a clinical grade PGx level of evidence. In our sample, 11% (*n* = 7/65), 63% (*n* = 41/65) and 26% (*n* = 17/65) were classified as normal, heterozygous, or homozygous, respectively. The frequency of the homozygotic genotype of rs1801133 (c.665C>T) was 35% (*n* = 6/17), while that of the homozygotic genotype of rs1801131 (c.1286A>C) was 65% (*n* = 11/17). The frequency of *MTHFR* genotypes stratified by age is shown in Appendix A.

### 3.3. Clinical Severity and MTHFR rs1801133 and rs1801131 Genotypes

Using the RTT-anchored CGI-S, we examined the symptom severity profile across the clinical domains for *MTHFR* rs1801133 and rs1801131 genotypes. This showed that the clinical severity profiles were different between the *MTHFR* homozygous and heterozygous genotypes (Table 2A). Varying patterns were noted across the clinical domains. A wider spectrum of severity was seen in the heterozygous individuals, whereas homozygous individuals were noted to have a narrower distribution for language and communication, ambulation, hand-use and eye contact clinical domains. The greatest difference in ambulation between the homozygous and heterozygous genotypes were noted at CGI-S:6 (60% homozygous vs. 31.8% heterozygous). Similarly, when looking at the clinical domain for hand use, a greater percentage of heterozygous individuals were concentrated at CGI-S:4 (45.4%), whereas homozygous individuals were more concentrated at CGI-S:6 (35.2% homozygous vs. 11.3% heterozygous). The clinical domain for autonomic symptoms showed a greater percentage of heterozygous individuals in CGI-S:4 (46.8% heterozygous vs. 17.6% homozygous). Interestingly, there was a greater percentage of homozygous individuals at CGI-S:6, which requires the presence of cyanosis (29.4% homozygous vs. 14.8% heterozygous). Furthermore, 21.2% of heterozygous individuals experienced no seizures compared to 5.8% of homozygous individuals. For the seizure clinical domain, the percentage of homozygous individuals was more concentrated at CGI-S:6 (35.2% homozygous) whereas the heterozygous individuals were more evenly distributed across all CGI-S ratings. Attentiveness profiles for homozygous and heterozygous individuals were similar, ranging between CGI-S:3 and CGI-S: 6 with the greatest difference noted at CGI-S: 5 (41.1% homozygous vs. 21.1% heterozygous). When examining overall RTT-anchored CGI-S ratings (Table 2B), the results indicated that individuals with a homozygous *MTHFR* genotype had significantly greater CGI-Severity scores than individuals with a non-homozygous *MTHFR* genotype (Z = −2.44, *p* = 0.015). When comparing the ratings of moderately impaired (4), markedly impaired (5), severely impaired (6) and extremely impaired (7), individuals with the homozygous *MTHFR* genotype were more impaired than those in the non-homozygous *MTHFR* group (Z = −2.06, *p* = 0.039).

### 3.4. AED Use and MTHFR Genotypes in RTT Patients

The use of the following AEDs was assessed between the homozygous and non-homozygous genotypes: Carbamazepine, Brivaracetam, Clobazam, Clonazepam, Lacosamide, Lamotrigine, Levetiracetam, Midazolam, Oxcarbazepine, Sodium Valproate, Topiramate and Zonisamide. An independent sample Mann–Whitney U test indicated no statistically significant difference between the number of AEDs prescribed for individuals in the homozygous *MTHFR* group when compared to individuals in the non-homozygous *MTHFR* group (Z = −0.56, *p* = 0.573) (Table 3). The frequency of the different AEDs used between the homozygous and the non-homozygous *MTHFR* group is presented in Appendix A.

### 3.5. RTT Diagnoses Categorised by MTHFR rs1801133 and rs1801131 Genotypes

Individuals with RTT were categorised according to the *MTHFR* rs1801133 and rs1801131 genotypes. There was no pattern between the mutation profile of RTT patients and the co-occurring *MTHFR* rs1801133 and rs1801131 polymorphisms tested (Appendix A).

## 4. Discussion

Our study, for the first time, has (I) identified and reported upon the frequency of co-occurring *MTHFR* rs1801133 and rs1801131 genotypes in patients with RTT and (II) demonstrated that those RTT patients with the homozygous *MTHFR* genotype have an overall worse symptom profile when compared to non-homozygous individuals as determined by the RTT-anchored CGI-S. This suggests that in a subset of RTT patients, co-occurring *MTHFR* rs1801133 and rs1801131 genotypes may act as associative genetic modifiers of clinical severity. However, this finding should be tempered because homocysteine and folate levels were not reported for our sample and the phenotypical correlates of individuals with *MTHFR* homozygosity are therefore of an associative level rather than a certainty of findings. Our follow-up work intends to measure homocysteine and folate levels from our patient group to explore the certainty of our findings and further examine how *MTHFR* rs1801133 and rs1801131 genotypes may act as associative genetic modifiers of clinical severity in RTT. Notwithstanding this limitation, this is the first study that has profiled *MTHFR* polymorphisms and its association with symptom severity in a paediatric neurodevelopmental disorder, in this case RTT, as part of routine clinical care in a National and Specialist CAMHS in the UK.

Individuals with the heterozygous or homozygous c.665C>T genotype have enzymatic activity 67–65% and 30–25% compared to the normal phenotype, respectively [12,37]. For those with c.1286A>C, the MTHFR activity is 83% and 61% for heterozygous and homozygous genotypes when compared to those with normal MTHFR activity [14]. The MTHFR c.665C>T and c.1286A>C genotypes vary across populations and depend on ethnicity [38]. In one study of 1405 subjects in the US, the highest c.665C>T variant frequency was found in Hispanics and Caucasians [39]. This aligns with the data from the 1000 Genomes projects that showed the highest frequency of c.665C>T in Hispanics (47%), Europeans (36%) and South Asians (12%) [39,40]. In comparison, the frequency of c.1286A>C is the highest in Southeast Asians (42%) when compared to Europeans (31%) [39,40]. Other evidence has shown that the homozygosity of the c.665C>T genotype was 18%, while that of the homozygous c.1286A>C genotype was 12.5% [41]. In our total sample, 63% of the genotypes were heterozygous while 26% were homozygous. Of those showing homozygosity, 35% of the genotypes were homozygous for c.665C>T and 65% were homozygous for c.1286A>C. Caution is needed when extrapolating *MTHFR* rs1801133 and rs1801131 genotype frequencies from studies undertaken in other populations using neurotypical subjects [38,39,40,41] to the observed *MTHFR* frequencies in our sample.

It has been suggested that some AEDs such as oxcarbazepine and topiramate might exacerbate *MTHFR* deficiency symptoms by worsening remethylation [42]. We, therefore, wanted to assess whether there is a higher frequency of AED use in the homozygous *MTHFR* group. Our data showed no difference in the number of AEDs used between the homozygous group compared to those in the non-homozygous *MTHFR* group.

Both *MTHFR* and *MECP2* have a function in methylation. However, it is unknown how impaired remethylation of homocysteine to methionine caused by reduced 5-MTHF underpins the global methylation changes regulated by pathogenic *MECP2*. Although our current findings are at an associate level, i.e., patients with the homozygous *MTHFR* genotype may synergistically contribute to the phenotype of patients with RTT, it was more than two decades ago when it was first surmised that low levels of 5-MTHF in Rett patients reduce folate transport in the CNS [43]. It was shown that the levels of 5-MTHF can be normalised with improved clinical outcomes following oral folinic acid supplementation. Following that work, there is further evidence to suggest folate receptor autoimmunity exists in RTT patients [44]. Low 5-MTHF was also said to be associated with epilepsy [45]. In that study, improvements in hand stereotypies were also observed following 1 year after folinic acid supplementation [45]. However, another study that assessed 5-MTHF levels in the cerebrospinal fluid (CSF) of 76 females with RTT showed no clinically significant changes in spinal 5-MTHF levels [46]. In young infants with severe *MTHFR* deficiency, early betaine treatment was suggested to prevent developmental psychomotor decline [47]. However, there have also been a few cases of brain oedema in patients with *MTHFR* deficiency treated with betaine [24]. Some others have suggested that folinic acid does not improve the clinical course of *MTHFR* deficiency [25]. When looking through the lens of neuropsychiatric disorders, folate supplements can improve clinical outcomes. A systematic review that looked at the efficacy of folate supplements in 23 studies of psychiatric disorders demonstrated that levomefolic acid or 5-methylfolate could improve clinical outcomes for patients with some psychiatric disorders [48]. However, neither betaine nor folate seem to objectively improve clinical outcomes in RTT patients, although parent-reported measures do show improvement in some clinical domains such as sleep, breathing issues, hand stereotypies, ambulation, and communication in children less than 5 years old [49]. Similarly, although folinic acid supplementation raises 5-MTHF in RTT patients, there are no significant improvements in clinical outcomes [50,51].

Further work would be needed to determine whether the MTHFR remethylation pathway is disrupted in RTT patients. However, the impaired dysregulation of methylation markers involved in the MTHFR pathway in RTT may support this notion [52]. Pre-emptive PGx testing to stratify RTT population groups for *MTHFR* genotypes could help to identify the appropriate patient subsets. As previously indicated, individuals with homozygotic genotypes should have their plasma methionine and plasma total homocysteine levels assessed. It would be prudent to perform these investigations as early as possible to see if there is a dysfunction of the remethylation pathway. Evidence indicates that early initiation of betaine or folate supplementation can improve neurodevelopmental outcomes in remethylation disorders [47,53,54]. Indeed, data from the European Network and Registry for Homocystinurias and Methylation Defects (E-HOD) registry demonstrated that pre-symptomatic diagnosis was a predictive factor for better neurodevelopment outcomes in patients with *MTHFR* deficiency [55].

Previous evidence has suggested genotype–phenotype correlations in RTT and some RTT variants are thought to have a more severe phenotype than others [56]. Clustering of RTT patients according to co-occurring *MTHFR* genotypes identified from PGx testing in our study did not reveal patterns concerning the prevalence of a particular type of RTT mutation. In a previous study examining the treatment of folate and betaine in RTT, although there were some differences in RTT variants between the placebo and treatment arms, there was insufficient statistical power to determine the impact of these variants [49]. In our study, the numbers are too small to examine whether a particular pathogenic RTT variant in the presence of co-occurring *MTHFR* polymorphism would have more impact on clinical severity. A larger sample would be required for a definitive conclusion; however, given the rare nature of RTT, it may not be possible to obtain sufficient numbers. Even in cases where genotype–phenotype relationships in RTT have been suggested, the clinical presentation can still be highly variable [57,58] and recent expert consensus guidelines for the management of RTT have also indicated that genotype–phenotype correlations in RTT are inaccurate [59]. The diagnosis of RTT is based on clinical criteria that are independent of molecular profiles [34,59]. Therefore, understanding genotype–phenotype correlations in the presence of co-occurring *MTHFR* polymorphism in RTT may not add value to the evidence base. It is likely that methylation would be the most modified in those with RTT and co-occurring *MTHFR* deficiency and this combination could have the greatest impact on neurodevelopmental outcomes (Figure 1).

## 5. Conclusions

While the phenotypical correlates of subjects with *MTHFR* homozygosity remain to be established, our findings do point towards an association between *MTHFR* rs1801133 and rs1801131 homozygosity and symptom profiles in RTT. This could be relevant for the most vulnerable patients who are at higher risk. However, homocysteine and folate levels were not measured, and therefore the findings of our study are of an associative nature. It would be interesting to investigate how these genotypes affect the concentration of homocysteine or folate levels in our sample. Age-related effects of PGx are important. Studies on paediatric PGx-guided prescribing are scarce, and three drugs (codeine, lansoprazole, and omeprazole) are suggested to have age-specific paediatric guidance [60]. Despite this, others have shown that PGx-guided treatment would be helpful for mood disorders and gastritis/esophagitis in the paediatric population [61]. Other exemplars of paediatric PGx implementation also exist [62].

As we move forward, genome-wide association studies (GWASs) could help in further confirming PGx-predisposing loci, especially in the case of *MTHFR* deficiency. However, only about 10% of GWASs have evaluated drug responses [63]. If there are rare variations of specific pharmacogenes, a GWAS may not have sufficient power to detect an association relevant to PGx guidance that will be clinically actionable [64]. Polygenic risk scoring might also help predict an individual’s response to medication [62]; however, in RTT, the small sample size would probably limit the ability to obtain accurate polygenic risk scores [65]. An alternative strategy would be to use pharmacoepigenomics to show how epigenetic modifications, i.e., methylation, have a role in treatment responses such as in the effectiveness of antidepressants and/or antidepressant response [66]. This strategy has shown some promise as hypomethylation in LINE-1 is suggested to be associated with poor response to risperidone in psychosis [67].

### Limitations 

Rare genetic variants of *MTHFR* may be present in our sample. Nevertheless, these rare variants would not necessarily be detected by PGx testing, and therefore more complex associations between MTHFR remethylation impairments and RTT could be missed. We also did not directly measure remethylation impairments reflected by the methylation markers such as 5-MTHF, methionine, S-adenosylmethionine (SAM); S-adenosylhomocysteine (SAH), which are suggested to have an important role in methylation in RTT [52] or folate levels. Therefore, the phenotypical correlates of the subjects with *MTHFR* homozygosity in our sample should be treated with caution.

A recent review has indicated bias for including Europeans in PGx research [68]. In a scoping review of 23,701 articles that searched for the terms “precision medicine” or “pharmacogenetics”, less than 5% stated the ethnicity of the sample [69]. The lack of PGx studies in ethnically diverse populations underscores the notion that the association between *MTHFR* rs1801133 and rs1801131 homozygosity and symptom profiles in RTT in our sample would not necessarily be reflective of RTT patients in other geographical regions. Future studies will need to implement PGx testing in diverse ethnic groups not only to understand different variant frequencies within these groups and their clinical implications but also to address gaps and health inequalities in PGx research.

## Figures and Tables

**Figure 1 brainsci-14-00624-f001:**
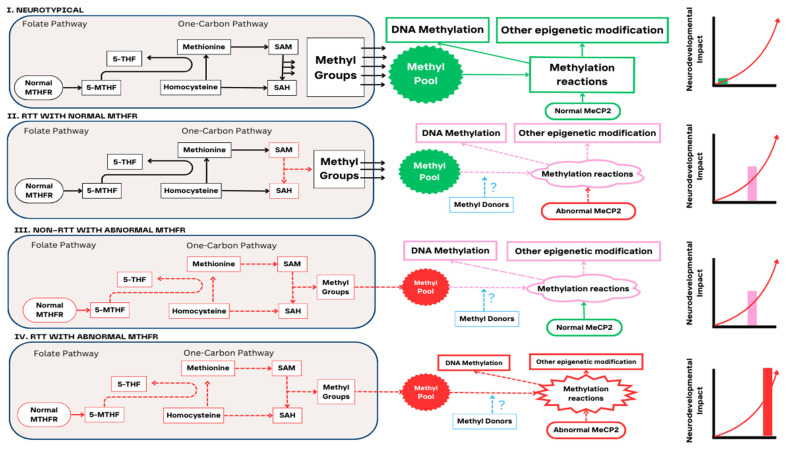
Neurodevelopmental impact and methylation pathways. Abbreviations: 5-THF (5-tetrahydrofolate); *MECP2* (methyl-CpG-binding protein 2 [MeCP2]) gene; MTHFR (methylenetetrahydrofolate reductase); RTT (Rett Syndrome); SAH (S-adenosylhomocysteine); SAM (S-adenosylmethionine). Methylation is expected to be normal in neurotypical individuals (I) and the neurodevelopmental outcomes are as observed in the general population. In individuals with RTT, methylation is disrupted due to a pathogenic *MECP2* variant and possibly due to impaired methylation markers and these impairments result in neurological dysfunction (II). In those with abnormal *MTHFR*, methylation is altered, and neurodevelopmental outcomes are also affected (III). Methylation is the most disrupted in those individuals with RTT and probably co-occurring *MTHFR* homozygosity (IV). This genotype may have the greatest impact on neurodevelopmental outcomes.

**Table 1 brainsci-14-00624-t001:** Sample (*n* = 65) descriptives. (**A**) Subject characteristics. (**B**) *MTHFR* rs1801133 and rs1801131 genotypes and predicted impact on MTHFR activity in 65 patients with RTT.

**(A)**				
			**Patients with a PGx Test (*n* = 65)**	
Diagnoses				
Rett Syndrome			60	
Atypical Rett Syndrome			5	
Ethnicity				
White: English, Welsh, Scottish, Northern Irish or British			53	
Mixed/Multiple ethnic groups			8	
Black/African/Caribbean/Black British			1	
Asian/Asian British			2	
Other ethnic groups			1	
Age (years)				
Mean (min. max)			18.7 (2.5; 47.5)	
Standard deviation			±12.1	
**(B)**				
***MTHFR*** **Genotypes**		**Phenotype (MTHFR Activity) ***		** *n* **
**rs1801133** **(c.665 C>T)**	**rs1801131** **(c.1286 A>C)**			
AG	GT	Mild		10
AG	TT	Partial		15
GA	GT	Partial		1
GA	TT	Partial		2
GG	GT	Partial		13
TT		Yes (reduced)		1
AA	TT	Yes (reduced)		5
GG	GG	Yes (reduced)		11
CC		Normal		1
GG	TT	Normal		6

* Predicted MTHFR activity (mild, partial, or reduced activity) as described in the individual patient pharmacogenomic reports for 65 patients. Notes: [I]. The normal designation was given for those that were predicted to have normal MTHFR activity (*n* = 7). [II]. The heterozygous designation was given for those predicted to have either mild or partial MTHFR activity. Individuals who had the slightly reduced MTHFR phenotype were also categorized as partial (*n* = 41). [III]. The non-homozygous (rs1801133 and rs1801131) descriptor was given to those predicted to have MTHFR activity as assigned for the normal and heterozygous groups (*n* = 48). [IV]. The homozygous (rs1801133 and rs1801131) descriptor was given for those with the most reduced MTHFR activity (*n* = 17).

**Table 2 brainsci-14-00624-t002:** RTT Clinical Global Impression Scale Severity (CGI-S) ratings between non-homozygous and homozygous *MTHFR* rs1801133 and rs1801131 genotypes in RTT patients. (**A**) Number of patients (%) across different CGI-S clinical domains for *MTHFR* rs1801133 and rs1801131 genotypes. (**B**) Overall RTT-anchored CGI-S ratings.

**(A)**							
**CLINICAL** **DOMAINS**	**CGI-S: 1**	**CGI-S: 2**	**CGI-S: 3**	**CGI-S: 4**	**CGI-S: 5**	**CGI-S: 6**	**CGI-S: 7**
Language/ Communication	(CSS = 0)	(CSS < 5)	(CSS 5–10)	(CSS 10–20)	(CSS 20–25)	(CSS 25–35)	(CSS 35–40)
	Normal	Appropriate. May have unusual features such as perseveration/echolalia. Reading disability/dyslexia	Phrase-sentences. May have conversations or echolalia	Words (<5). Babbles. Makes choices. 25–50%	No words. Babbles. Makes choices ≤25%	Vocalizations. Occasionally screams. Makes no choices or only rarely makes choices	No words. No vocalizations. Screams. No choices
Homozygous *n* = 16	0 (0%)	0 (0%)	0 (0%)	5 (31.2%)	5 (31.2%)	6 (37.5%)	0 (0%)
Heterozygous *n* = 43	0 (0%)	0 (0%)	2 (4.65%)	11 (25.5%)	13 (30.2%)	15 (34.8%)	2 (4.6%)
Ambulation	No impairment	Normal, may have slight evidence of dystonia/ataxia/dyspraxia upon careful examination	Walks, able to use stairs/run. May ride tricycle or climb	Walks independently, unable to use stairs or run	Walks with assistance	Stands with support or independently. May walk with support. Sits independently or with support	Cannot sit. Does not stand or walk
Homozygous *n* = 15	0 (0%)	0 (0%)	0 (0%)	2 (13.3%)	2 (13.3%)	9 (60%)	2 (13.3%)
Heterozygous *n* = 44	0 (0%)	1 (2.27%)	6 (13.6%)	7 (15.9%)	12 (27.2%)	14 (31.8%)	4 (9.1%)
Hand Use	Completely normal, no impairment	Normal, may have slight fine motor issues	Bilateral pincer grasp. May use pen to write but has some fine motor issues like tremor	Reaches for objects, raking grasp or unilateral pincer. May use utensils/cup	Reaches. No grasps.	Rarely occasionally reaches out. No grasp	None
Homozygous *n* = 17	0 (0%)	0 (0%)	0 (0%)	5 (29.4%)	5 (29.4%)	6 (35.2%)	1 (5.88%)
Heterozygous *n* = 44	1 (2.27%)	0 (0%)	2 (4.55%)	20 (45.4%)	15 (34.1%)	5 (11.3%)	1 (2.27%)
Social (Eye Contact)	Normal	Occasional eye gaze avoidance	Appropriate eye contact, >30 s	Eye contact <20 s	Eye contact <10 s	Eye contact, inconsistent 5 s	No eye contact
Homozygous *n* = 14	0 (0%)	0 (0%)	1 (7.14%)	7 (50%)	6 (42.8%)	0 (0%)	0 (0%)
Heterozygous *n* = 39	0 (0%)	5 (12.8%)	9 (23.1%	12 (30.7%)	10 (25.6%)	3 (7.69%)	0 (0%)
Autonomic	None	Minimal	No or minimal breathing abnormalities (<5% of times observed) and warm, pink extremities	Breathing dysrhythmia <50%. No cyanosis. Cool UE and LE pink	Breathing dysrhythmia 50%. No cyanosis. Cool UE and LE pink	Breathing dysrhythmia, 50–100%, maybe with cyanosis. Cold LE or UE, may be blue	Breathing dysrhythmia, constantly with cyanosis. Cold UE and LE. Mottled/blue
Homozygous *n* = 17	0 (0%)	1 5.88%)	4 (23.5%)	3 (17.6%)	4 (23.5%)	5 (29.4%)	0 (0%)
Heterozygous *n* = 47	0 (0%)	0 (0%)	6 (12.7%)	22 (46.8%)	12 (25.5%)	7 (14.8%)	0 (0%)
Seizures	None	None or controlled	None, with or without meds	Monthly– Weekly	Weekly	Weekly–Daily	Daily
Homozygous *n*= 17	1 (5.8%)	3 (17.6%)	1 (5.88%)	4 (23.5%)	0 (0%)	6 (35.2%)	2 (11.7%)
Heterozygous *n* = 47	10 (21.2%)	8 (17.0%)	5 (10.6%)	9 (19.1%)	5 (10.6%)	6 (12.7%)	4 (8.51%)
Attentiveness	Entirely normal	Occasional inattention	Attentive to conversation and follows commands	50–100% of the time	50% of the time	Less than 50% of the time	0%
Homozygous *n* = 17	0 (0%)	0 (0%)	2 (11.7%)	4 (23.5%)	7 (41.1%)	4 (23.5%)	0 (0%)
Heterozygous *n* = 42	0 (0%)	0 (0%)	8 (19.0%)	11 (26.1%)	11 (26.1%)	12 (28.5%)	0 (0%)
**(B)**							
**RTT CGI-S** **Clinical Domains**			***MTHFR*** **rs1801133 and rs1801131 Genotypes**				
			**Non-Homozygous** **(*n* = 47 *)**			**Homozygous** **(*n* = 17)**	
Normal * (1)			0			0	
Borderline impaired (2)			0			0	
Mildly impaired (3)			4			0	
Moderately impaired (4)			21			3	
Markedly impaired (5)			16			10	
Severely impaired (6)			6			4	
Extremely impaired (7)			0			0	

The difference in the number of patients in the clinical domains indicates that there was not sufficient clinical information available to complete the scoring. The colours of the clinical domains indicate increasing clinical severity. RTT-CGI-S provided in the work of Neul JL, Glaze DG, Percy AK, Feyma T and Beisang A. et al. entitled Improving Treatment Trial Outcomes for Rett Syndrome: The Development of Rett-specific Anchors for the Clinical Global Impression Scale. J Child Neurol. 2015 Nov;30(13):1743–8. reference [36]. * One individual did not have sufficient clinical information to complete the RTT CGI-S.

**Table 3 brainsci-14-00624-t003:** Frequency of AED use and *MTHFR* rs1801133 and rs1801131 genotypes in RTT patients (*n* = 65).

Number of AEDs *	*MTHFR* rs1801133 and rs1801131 Genotypes			
	Homozygous (*n* = 17)	Homozygous (%)	Non-Homozygous (*n* = 48)	Non-Homozygous (%)
None	5	29%	15	31%
1	7	41%	24	50%
2	4	24%	7	15%
3	1	6%	1	2%
4	0	0%	0	0%
5	0	0%	1	2%
Average number of AEDs used	1.1 ± 0.9	100%	1.0 ± 0.9	100%

* The AEDs prescribed to each patient at the time of the pharmacogenomics test.

## Data Availability

Data can be obtained upon reasonable request from the corresponding author.

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
