# Peer review of "Co-Occurring Methylenetetrahydrofolate Reductase (MTHFR) rs1801133 and rs1801131 Genotypes as Associative Genetic Modifiers of Clinical Severity in Rett Syndrome"

_brainsci, 2024, doi:10.3390/brainsci14070624_

Round 1

Reviewer 1 Report

Comments and Suggestions for Authors

The manuscript reports a survey of co- Methylenetetrahydrofolate Reductase (MTHFR) polymorphisms and their phenotypical correlates in Rett syndrome (RTT).  The study included 65 subjects (2.5 to 47.5 years), 17 of them with a mutant homozygous MTHFR genotype.  Mainly using the CGI-S with anchors for RTT the authors found that, compared with those without the homozygous MTHFR genotype, the group carrying the polymorphism had worse motor symptoms and higher seizure frequency.  This apparently opportunity study raises several important issues, including the value of pharmacogenomic assessments and the need for identifying genetic modifiers of clinical severity in RTT and other genetic neurodevelopmental disorders.  However, although the reported findings are intriguing, the manuscript presents are multiple conceptual and methodological concerns listed below:

1. Levels of homocysteine are not reported; therefore, the phenotypical correlates of the subjects with MTHFR homozygosity are speculative.  Moreover, clinical severity of MTHFR polymorphisms and therapeutic correction depend also on folate levels which are not provided either.

2. Further supporting the concerns about overinterpretation of the MTHFR polymorphism data are the fact that increases in homocysteine tend to be mild and the corresponding guidelines of the American College of Medical Genetics and Genomics (ACMG): ACMG Practice Guideline: lack of evidence for MTHFR polymorphism testing. Genetics in Medicine. 2013;15(4):153-6.  Evidence of association between MTHFR polymorphisms and severe developmental phenotype is weak and, again, speculative without knowledge on levels of homocysteine and folate at critical developmental periods.

3. It is unclear how the CGI-S scale was used in the study.  The 7 RTT-specific anchors should be used as the basis for a single CGI (Global) score.  The CGI and its anchors are not Likert scales. 

4. Given the variety of seizures in RTT, including relatively milder generalized seizures (i.e., atypical absence) report of frequency is insufficient.   

5. It is unclear why is the number of subjects for the CGI-S evaluations lower than for the “clinical domains”.

In addition to these major concerns, the manuscript presents other deficiencies, such as unclear statements or awkward grammar (meaning of “Rett Syndrome presents with different symptom profiles and patients can become treatment non-responders”, a statement without references). The and inaccurate placement of references (reference #26 and not #25 is the right for the RTT CGI). 

Comments on the Quality of English Language

Reviewer 2 Report

Comments and Suggestions for Authors

RTT pts have been reported to be at higher risk of adverse drug reactions (ADRs). The current Ms focuses on the possible role of co-occurring remethylation disorders (MTHFR homozygous mutant genotype) in 65 pts affected by RTT. The MTHFR genotype was tested by use of pharmacogenomic (PGx) testing and clinical severity was rated by a subjective clinical scale (CGI). A co-occurring mutant homozygous MTHFR genotype was associated with worse CGI ratings, less likely ambulation, worse hand use, and higher seizure frequency. The AA conclude that a co-occurring mutant homozygous MTHFR genotype “may worsen motor symptoms and could be associated with increased seizure frequency in a subset of patients”.

 The rationale is of interest. The evidence is somewhat suggestive for the hypothesis formulated by the AA. However, a number of pitfalls indicate the need for a major revision.  

1) Clinical severity rating of RTT can be improved. Did the AA use other severity scale for the disease ?  Is severity assessment perfomed at the moment of sample collection ?

2) No data regarding the pro kg dose of AEDs are reported. Please provide and discuss.

3) “…Samples were obtained from buccal swabs between April 2021 and January 2024” and ethical approval date is 19 October 2022. Please clarify

4) Discussion is quite confusing and needs to be shortened and should be extensively rephrased.

5) The MTHFR mutation also cause issues with detoxing properly, as well as issues with hormone balance and immune system function. It has been associated with autoimmune conditions such as fibromyalgia, Hashimoto's, and systemic lupus erythematosus (SLE). On the other hand, polymorphisms in MECP2 have been linked to increased susceptibility to autoimmune diseases in humans, such as SLE, thyroid diseasesand primary Sjogren’s syndrome. Did the AA find some evidence in the clinical features of the enrolled patients ? I would suggest to explore also this aspect.

Minor comments

1) Please provide information about the facilities Admera Health LLC and Innovative Gx Laboratories

Comments on the Quality of English Language

The Ms Language style needs major editing.

Reviewer 3 Report

Comments and Suggestions for Authors

The manuscript entitled “Co-occurring Methylenetetrahydrofolate Reductase (MTHFR) Deficiency in Rett Syndrome: A Possible Link to Motor Impairments” examines the possible link between MTHFR genotypes and motor impairments accompanying Rett Syndrome rather than the link between MTHFR deficiency and motor symptoms. In this regard, the title of the study has to be changed, since the study does not assess MTHFR deficiency.

In addition, several issues need to be addressed.

1.     Some information is lacking in the Abstract, i.e. mean age, the designation of MTHFR SNPs. Please, also indicate in Abstract and throughout the text a widely used form of SNP names based on dbSNP (https://www.ncbi.nlm.nih.gov/snp).

2.     What was a rationale for selection of certain MTHFR variants for the analysis? It has to be mentioned in the Introduction with references for previous research.

3.     The authors reported that they used a panel of 67 genes for genetic screening, why did they use only two SNPs in the MTHFR gene?

4.     The authors reported the use of t-tests for statistical analysis. However, this statistical criterion is used for normally distributed quantitative dependent variable (i.e. CGI-S clinical domains, the CGI-S average, the overall CGI-S, and AED use). Since the sample size is rather small, it is unlikely that quantitative variables were normally distributed; therefore, statistical analysis has to be based on non-parametric criteria.

5.     Table 1B states the information regarding MTHFR activity. It remains unclear what was a method for measuring this phenotype. Moreover, it is unappropriate to assign normal, heterozygous, and homozygous mutant based on the MTHFR activity rather than on genotypes themselves (Supp. Information 2). This figure has to be revised.

6.     What is a basis to assign a “mutant” genotype? Please, indicate the functional studies conducted on animal models or cell lines, which indicate the functional significance of examined SNPs on hyperhomocysteinemia. In addition, it is required to report the change in aminoacids caused by nucleotide substitution change as well as population frequency data to make the assignment of “mutant” allele. Otherwise, it is inappropriate to assign some of the allelic variants as mutant ones. I think it is more appropriate to designate these SNPs as genetic variants or polymorphisms rather than mutations. Please, revise the manuscript for the correct use of term “mutant”.

7.     In addition, the sentence in subsection 3.2 has to be clarified, i.e. “The PGx panel reported on the two MTHFR variants reported as having enzymatic deficiency: c.665C>T and c.1298A>C.” The authors just report the designations of examined SNPs without indication of “risky” alleles. Such changes have to be made throughout the text.

8.     It is not clear, what was the assignment to “mutant homozygous” and “not homozygous” groups? Does “mutant homozygous” require the presence of homozygotic genotype of both SNPs simultaneously? Please, give more detailed explanation.

9.     It is not clear, what was a sense to include AEDs, which were non-administered to individuals examined in the present study (Supp. Information 3)? I suggest excluding them from the table.

10.  Conclusion is too large, transfer some part of it into Discussion. In turn, Discussion has to be revised on the basis of redone correct statistical analysis and correct grouping of genotypes of examined SNPs.

11.  I suggest to rephrase the aliases for ethnicity (i.e. Table 1 and throughout the text). What is white, white-other, black? Please, report widely used ethnic description. I also think it is appropriate to delete the row “gender”, since all the examined subjects were women. Also check the age values (min, max, standard deviation) for their correctness.

12.  Indicate in Table 2 that statistically significant differences are marked in bold.

13.  There are some inconsistencies in the use of terms, i.e. the sentence in Discussion section “Individuals with the heterozygous or homozygous c.665C>T phenotype have enzymatic activity 67-65% and 30-25% of the normal phenotype, respectively [9, 27].” It seems that “phenotype” has to be changed into “genotype”.

Round 2

Reviewer 2 Report

Comments and Suggestions for Authors

The Manuscript is improved. The AA answered to the raised points.

Comments on the Quality of English Language

Minor language revision is needed 

Author Response

Dear Reviewer 2,

Thank you for your reply that the manuscript is improved and that we have answered all of the raised points.

The manuscript has been checked for English language and has been improved. These changes are provided in the manuscript as deletions and highlighted text.

Best wishes,

Dr Jatinder Singh

Reviewer 3 Report

Comments and Suggestions for Authors

The manuscript has been modified, all the issues have been addressed. The manuscript can be published in the present form. 

Author Response

Dear Reviewer 3,

Thank you for your reply that the manuscript has been modified, all the issues have been addressed and that the manuscript can be published in the present form.

Best wishes,

Dr Jatinder Singh
